# Pathogenicity Analyses of Rice Blast Fungus (*Pyricularia oryzae*) from *Japonica* Rice Area of Northeast China

**DOI:** 10.3390/pathogens13030211

**Published:** 2024-02-28

**Authors:** Dongyuan Wang, Feng Zhu, Jichun Wang, Hongguang Ju, Yongfeng Yan, Shanyan Qi, Yuping Ou, Chengli Tian

**Affiliations:** 1Plant Pathology Division, Institute of Plant Protection, Jilin Academy of Agricultural Sciences, Changchun 130033, China; wangdongyuan1998@yeah.net (D.W.); zhufeng0726@163.com (F.Z.); tianchengliqq@163.com (C.T.); 2Plant Pathology Division, College of Plant Protection, Jilin Agricultural University, Changchun 130118, China; 3Rice Breeding Division, College of Agriculture, Yanbian University, Yanji 130002, China; 4Rice Breeding Division, Institute of Rice Research, Jilin Academy of Agricultural Sciences, Changchun 130033, China; yanyongfeng@126.com

**Keywords:** rice blast, *Pyricularia oryzae*, pathogenicity, *Japonica* rice, northeast China

## Abstract

In order to understand the pathogenicity differentiation of rice blast fungus (*Pyricularia oryzae* Cavara), a total of 206 isolates of *P. oryzae* were collected from three Japonica rice regions in Jilin Province, northeast China. Pathogenicity test showed that the reaction pattern of 25 monogenic differential varieties (MDVs) of rice (*Oryza sativa* L.) demonstrated a wide pathogenic diversity among the isolates. Those MDVs harbor 23 resistance (*R*) genes with the susceptible variety Lijiangxintuanheigu (LTH) as control. Virulent isolates of MDVs harboring *R* genes *Pish*, *Pit*, *Pia*, *Pii*, *Pik-s*, *Pik*, *Pita* (two lines), and *Pita-2* (two lines) had high frequencies ranging from 80 to 100%, to MDVs harboring *R* genes *Pib*, *Pi5*(t), *Pik-m*, *Pi1*, *Pik-h*, *Pik-p*, *Pi7*(t), *Piz*, *Piz-5*, and *Piz-t* showed intermediate frequencies ranging from 40 to 80%, and to MDVs with *R* genes *Pi3*, *Pi9*(t), *Pi12*(t), *Pi19*(t) and *Pi20*(t) presented low frequencies ranging only from 0 to 40%. The U-i-k-z-ta pattern of race-named criteria categorized the 206 isolates into 175 races. Sub-unit U73 for *Pib*, i7 for *Pi3* and *Pi5*(t), k177 for *Pik-m*/*Pik-h*/*Pik-p*, z17 for *Pi9*(t), and ta332 for *Pi20*(t) were crucial on pathogenic differences in regions. Twenty-seven standard differential blast isolates (SDBIs) were selected to characterize resistance in rice accessions. This study could help to build a durable identification system against blast in the *Japonica* rice area of northeast China and enhance our understanding of the differentiation and diversity of blast races in the world.

## 1. Introduction

Blast, caused by the fungus *Pyricularia oryzae* Cavara (formerly *Magnaporthe oryzae* B. C. Couch), is the most damaging disease in rice worldwide [1]. The interaction between host resistance and fungus virulence in the pathosystem can be explained by a “gene for gene” theory: for every resistance gene in the host variety, there is a corresponding avirulence gene in the blast pathogen [2,3]. Based on the theory, a large number of differential varieties (DVs) with special resistance genes have been developed [4,5,6,7,8,9,10,11,12]. Since 2000, two sets of DVs originated from *Japonica* Lijiangxintuanheigu (LTH) and *Indica* CO39 have been reported [13,14]. The *Japonica* monogenic DVs are produced by introducing single resistance genes into a Chinese Japonica cultivar LTH one by one. The introduced *R* genes include *Pib*, *Pit*, *Pia*, *Pii*, *Pi3*, *Pi5*(t), *Pik-s*, *Pik-m*, *Pi1*, *Pik-h*, *Pik*, *Pik-p*, *Pi7*(t), *Pi9*(t), *Piz*, *Piz-5*, *Piz-t*, *Pita-2* (two lines), *Pita* (two lines), *Pi12*(t), *Pi19*(t), and *Pi20*(t). These DVs are the latest and have been broadly utilized in many countries or regions in the world, including China [15], Japan [16], Kenya [17], and Indonesia [18].

Blast isolates have played a critical role in the identification of resistant germplasm in rice breeding. Each standard differential blast isolate (SDBI) has special avirulent characteristics, so SDBIs have been developed and broadly used in resistance breeding programs [13,18,19,20]. Pathogenically diverse isolates with different avirulence gene proportions have been reported in Cambodia [21], West Africa [22], Bangladesh [19], and Vietnam [20]. However, there is no report about SDBIs originating from the DVs in the *Japonica* rice production area of China.

There are two sets of DVs used successively in northeast China, where *Japonica* rice is produced [23]. The first set of DVs has been used since 1976, including 7 DVs of Tetep, Zhenlong13, Sifeng43, Dongnong363, Guandong51, Hejiang18, and LTH [5]. Because their genetic backgrounds on *R* genes are poorly understood, their identified race information has poorly helped blast-resistance breeding. The second set of DVs contains 25 monogenic differential varieties (MDVs), which harbor 23 resistance genes originating from *Japonica* LTH. Wang et al. [15] were the first to use this set of MDVs, then followed by others [17,18]. Nowadays, the second set of MDVs has been applied to evaluate the pathogenic diversity of isolates, exchange resistant germplasm, understand resistance genetics of germplasm by artificial inoculation, and guide resistance breeding. For example, the *Japonica* variety Qinglin511 is bred from Jiudao44 as a gene donor, which harbors the *R* gene *Piz,* and the local elite variety Jijing88 as a recurrent parent. Field selection in each generation is assisted by blast isolate JLavrPiz-20220817 as an SDBI. As a result, Qinglin511 is resistant to blast with high-yielding and super grain quality [24].

It is well known that planting blast-resistance varieties is the most economical, effectual, and environment-friendly method for controlling rice blast [1,2,25]. Blast-resistance breeding is based on a suitable understanding of the pathogenic characteristics of local pathogens. Therefore, our goal in this study was to comprehensively understand the pathogenic characteristics of rice blast isolates in Jilin Province so as to serve blast-resistance breeding. In northeast China, including Heilongjiang, Liaoning, and Jilin provinces, plus nearby Russia, Japan, and North Korea, *Japonica* rice is planted on five million hectares, where blast is the most severe disease [23,26]. Therefore, this study focused on rice blast isolates, aiming to (1) understand virulence diversity and isolate frequency in the *Japonica* rice area of northeast China; (2) clarify predominant pathogenic types; and (3) select SDBIs for screening accessions to a special pathogen.

## 2. Materials and Methods

### 2.1. P. oryzae Single Spore Isolation and Conservation

During 2019–2021, we collected *P. oryzae*-infected rice panicles and leaves from three regions in Jilin Province, China: the west drought region (Region I), including Song-Yuan and Bai-Cheng; the central semi-humid region (Region II), including Chang-Chun, Ji-Lin, and Si-Ping; and the east semi-mountain humid region (Region III), including Yan-Ji, Tong-Hua, and Liao-Yuan (Figure 1, Table 1). To obtain a single spore of *P. oryzae*, the collected samples were incubated on moist filter paper in a Petri dish for 24 h with the protocol described by Wang et al. [15]. The incubated samples were shaken off on water–agar medium at 25 °C for 24–36 h until single spores could be noticed with long hyphae under the optical microscope. Each single spore was picked up using small hooks and cultured in a PDA slant medium at 25 °C for 5–7 d. Then, a single colony was transferred to rice straw agar medium covered with filter paper of 9 cm in diameter at 25 °C for 10 d. Finally, these cultures of all single isolates carrying mycelia and conidia were stored at −20 °C in sterile glass vials after necessary drying.

**Note:** Three regions were classified based on the climate differentiation in Jilin Province, northeast China. The west drought region includes Bai-Cheng City and Song-Yuan City; the center semi-humid region, including Chang-Chun City, Ji-Lin City, and Si-Ping City; and the east semi-mountain humid region, including Yan-Ji City, Tong-Hua City, and Liao-Yuan City. Two groups occurred in all three regions, and two subgroups were divided into Ia and Ib in both the central and east regions.

**Table 1 pathogens-13-00211-t001:** Blast isolates and frequencies in three regions of Jilin Province, northeast China.

	Number of Blast Isolates and Frequency (%)			
	Cluster Group			
Region	Ia	Ib	II	Total
West drought region	6 (18.75)	0 (0.00)	26 (81.25)	32 (100.00)
Central semi-humid region	17 (20.99)	1 (1.23)	63 (77.78)	81 (100.00)
East semi-mountain humid region	24 (25.80)	1 (1.08)	68 (73.12)	93 (100.00)
Total	47 (22.82)	2 (0.97)	157 (76.21)	206 (100.00)

### 2.2. Differential Varieties and Growth Conditions

Our study used 25 rice monogenic lines with a highly susceptible check variety, Menggudao (CK). These monogenic differential varieties (MDVs) harbor 23 single *R* genes, respectively, including IRBLsh-B(*Pish*), IRBLt-K59(*Pit*), IRBLb-B(*Pib*), IRBLa-A(*Pia*), IRBL-F5(*Pii*), IRBL3-CP4(*Pi3*), IRBL5-M[*Pi5*(t)], IRBLks-F5(*Pik-s*), IRBLkm-Ts(*Pik-m*), IRBL1-CL(*Pi1*), IRBLkp-K60(*Pik-p*), IRBLk-k[LT](*Pik*), IRBLkh-K3[LT](*Pik-h*), IRBL7-M[*Pi7*(t), IRBL9-W[*Pi9*(t)], IRBLz-Fu(*Piz*), IRBLz5--CA(*Piz-5*), IRBLzt-T(*Piz*-t), IRBLta2-Re(*Pita-2*), IRBLta2-Pi(*Pita-2*), IRBL12-M[*Pi12*(t)], IRBLta-K1(*Pita*), IRBLta-CP1(*Pita*), IRBL19-A[*Pi19*(t)], IRBL20-IR24[*Pi20*(t)], and the backcross parent LTH [27,28,29]. Seeds of the monogenic lines and LTH were generously provided by Cailin Lei from the Institute of Crop Sciences, Chinese Academy of Agricultural Sciences (ICS-CAAS). LTH is a *Japonica*-type local variety from Yunnan Province, China, which is broadly susceptible to rice blast isolates. The Menggudao is used as a super-susceptible control variety or induce variety susceptible to blast disease in disease nurseries in China. Five seeds of each MDV and CK were planted in a plastic tray (68 × 34 × 7 cm) containing local clay soil that had been treated with 0.1% carbendazim fungicide for 3 d. The tray was kept at 20 °C for the night and 28 °C for the day for about four weeks until seedlings were at the 4th–5th leaf stage [15].

### 2.3. Inoculation and Evaluation

Each isolate was inoculated on an oat agar medium with a standard method [30]. After 2 weeks, each conidial suspension was standardized to a concentration of 1 × 10^5^ spores/mL, then 200 mL standardized spore suspension was sprayed to each tray (1 square meter) with seedlings at the 4th–5th leaf stage using an atomizer. The trays were kept in the dark for 16 h before being transferred to a greenhouse at 25–28 °C with 95% relative humidity. At 6–7 d after inoculation, blast reaction was classified on a scale of 0–5, where 0 means no evidence of infection; 1 means brown specks smaller than 0.5 mm, no sporulation; 2 means brown specks about 0.5–1.0 mm in diameter, no sporulation; 3 means roundish to elliptical lesion about 1–3 mm in diameter with a gray center surrounded by brown margins, lesions capable of sporulation; 4 means typical spindle-shaped lesions capable of sporulation, longer than 3 mm with necrotic gray centers and water-soaked or reddish brown margins, little or no coalescence of lesions; and 5 means the same as type 4 but with half of one or two leaf blades killed by the coalescence of lesions as described by Hayashi and Fukuta [31] using Menggudao as susceptible check. The inoculation and evaluation were performed twice sequentially for each rice line and each blast isolate, and the most severe lesion type on the leaf was recorded according to the compatible reaction type. Scales 0–2 were classified as R (Resistant) and 3–5 as S (Susceptible).

### 2.4. Characterization of the Blast Isolates

The R and S reaction patterns of each MDV to the blast isolates were characterized according to the U-i-k-z-ta pattern criteria and pathogenic proportion to MDVs [18]. The diversity in the collection was measured with Simpson’s diversity index [32]. The index values range from 0 to 1, where 0 presents no diversity and 1 maximum diversity [19].

### 2.5. Selection of Standard Blast Isolates

A compatible reaction pattern between MDVs and blast isolates was used to select standard blast isolates (SDBIs). An SDBI had no pathogenicity to one of the MDVs but had very strong pathogenicity to all other MDVs. Based on the gene-for-gene theory, there is a corresponding avirulence gene existing in isolates, which could be used to identify the genetic background of rice accessions resistant to *P. oryzae* according to the interaction-compatible phenotype. Furthermore, we selected some super-virulent isolates for screening super-resistant germplasm.

### 2.6. Dendrogram Construction and Cluster Analysis

Cluster analysis was performed via Ward’s [33] hierarchical method, based on the data of infection scores of the 25 DVs and LTH by blast isolates, in the MEGA11 computer program, aligned using clustalw implemented with the maximum likelihood method.

## 3. Results

### 3.1. Race Information by U-i-k-z-ta Pattern Criteria and Cluster Groups

We identified 206 blast isolates from collected samples originating in three ecosystems: 93 isolates from Region I—the west drought; 81 from Region II—the central semi-humid; and 32 from Region III—the semi-mountain humid (Table 1; Figure 1). In accordance with the race name criteria of the U-i-k-z-ta pattern, the 206 isolates were categorized into 175 races based on the phenotype data (Appendix A). Race U73-i5-k177-z11-ta330 contained five isolates and proved to be predominant among all races, followed by four isolates in race U53-i7-k177-z17-ta330, U73-i7-177-z17-ta733, and U73-i5-k103-z00-ta330, three isolates in race U53-i1-k101-z02-ta330, U73-i5-k177-z07-ta330, and U73-i7-k177-z17-ta731, and two isolates in 12 races of U53-i7-k177-z07-ta330, U53-i7-k175-z07-ta730, U53-i7-k177-z17-ta733, U73-i1-k101-z01-ta330, U73-i1-k101-z01-ta332, U73-i5-k101-z01-ta330, U73-i1-k101-z00-ta330, U73-i5-k177-z11-ta332, U73-i7-k177-z03-ta330, U73-i5-k177-z01-ta730, U73-i7-k177-z17-ta333, and U73-i7-k177-z17-ta732. Each of the remaining 156 races contained one of the remaining 156 isolates.

The 206 isolates from three regions were classified into two groups, and Group I was divided into subgroups Ia and Ib (Appendix A). Group I had 49 isolates, while Group II had 157 isolates. In Group I, Subgroup Ia had 47 isolates. (Table 1; Figure 1). It was obvious that Group II was the dominant among the three regions. Moreover, there were no isolates belonging to Subgroup Ib in the west region.

Among the three regions, the east region had the largest isolate number (93), followed by the central region (81) and the west region (32). Similarly, among cluster Group II, the east region had the largest isolate number (68), followed by the central region (63) and the west region (26), and the frequencies were 73.12%, 77.18%, and 81.25%, compared to Group I (Table 1). The details of pathogenic characteristics were presented in the dendrogram (Appendix A).

### 3.2. Frequencies of Isolate Virulence toward MDVs

Analyses of the virulence frequency of blast isolates to 25 MDVs and LTH in three regions showed that 5 MDVs (*Pi3*, *Pi9*(t), *Pi12*(t), *Pi19*(t), and *Pi20*(t)) had low frequency (<40%), 10 MDVs (*Pib*, *Pi5*(t), *Pik-m*, *Pi1*, *Pik-h*, *Pik-p*, *Pi7*(t), *Piz*, *Piz-5,* and *Piz-t*) had moderate virulence frequencies (40–80%), and 10 MDVs (*Pish*, *Pit*, *Pia*, *Pii*, *Pik-s*, *Pik*, *Pita* (two lines), and *Pita-2* (two lines)) had high virulence frequencies (>80%) (Figure 2a–d). Thirty-six isolates were avirulent to LTH, while check variety Menggudao was susceptible to all isolates. Isolate frequencies to MDVs were similar in three regions, but the MDVs harboring *Piz-5* and *Piz-t* had lower frequency in Region III of east semi-mountain than in Region I of west drought and Region II of central semi-humid (Figure 2a–c).

### 3.3. Subgroup Reaction Type of MDVs

Among 206 isolates, race reaction types were diverse. In MDVs Subgroup “U”, there were 21 reaction types (U01, U03, U11, U13, U21, U22, U23, U31, U32, U33, U40, U43, U52, U53, U61, U62, U63, U70, U71, U72, and U73). Among them, U73 had the most types of 100 (48.5%), which were virulent to four MDVs of *Pish, Pib, Pit, Pia,* and LTH. Subgroup U53 had 44 types (21.4%), which were virulent to three MDVs of *Pish*, *Pit*, *Pia*, and LTH but not virulent to *Pib*. The *avrPib* existed in many isolates (Table 2).

In MDVs Subgroup “i”, there were eight reaction types: i7, i6, i5, i4, i3, i2, i1, and i0. Among them, three were major types, with i5 having the dominant types of 65 (31.6%) virulent to *Pii* and *Pi5*(t), followed by i7, with 63 types (30.6%) virulent to all three MDVs of *Pii*, *Pi3*, and *Pi5*(t), and i1, with 45 types (21.8%) virulent to *Pii*. *Pii* was more susceptible to *Pi3* and *Pi5*(t) (Table 2).

MDVs Subgroup “k” had 45 reaction types. Among them, k177 was the predominant reaction type in Jilin Province, containing 85 isolates (41.4% of all 206 isolates), and was virulent to MDVs for *Pik-s*, *Pik-m*, *Pi1*, *Pik-h*, *Pik*, *Pik-p*, and *Pi7*(t) (Table 2).

In MDVs Subgroup “z”, z17 and z07 had more types than others, with 40 (19.4%) and 30 (14.6%), respectively. Both had different virulent and avirulent reactions to MDV for *Pi9*(t) and largely influenced the pathogenic diversity in “z” (Table 2).

MDVs Subgroup “ta” had 28 reaction types, including ta000, ta010, ta021, ta030, ta031, ta120, ta130, ta132, ta220, ta221, ta230, ta232, ta233, ta300, ta312, ta320, ta322, ta330, ta331, ta332, ta333, ta431, ta620, ta713, ta730, ta731, ta732, and ta733. Among them, ta330 was dominant with 87 isolates (42.2%), virulent to *Pita-2*(Pi), *Pita-2*(Re), *Pita*(K1), and *Pita*(CP1), but avirulent to *Pi12*(t), *Pi19*(t), and *Pi20*(t) (Table 2). The *avrPi12*(t), *avrPi19*(t), and *avrPi20*(t) were predominant avirulence genes in Jilin Province (Table 2).

Evidently, from the above MDVs subgroup results, U73, U53, i7, i5, i1, k177, and ta330 were predominant reaction types, and their frequency was more than 20%. Like z17 and z07 were affected by *Pi9*(t) reaction phenotype, ta330 was affected by *Pi12*(t), and all of them crucially influenced isolate characterization (Table 2).

### 3.4. Standard Differential Blast Isolates (SDBIs) Characterization

Based on the race type information, we selected 25 blast isolates as standard differential blast isolates (SDBIs) for screening resistance accessions (Table 3). These SDBIs differentiated the 23 kinds of resistance genes among the 25 DVs. Specifically, the strains of LY1, TH11, SP11, and SY15 corresponded to the broad-spectrum resistance genes of *Pi19*(t), *Pi2*0(t), *Pi12*(t), and *Pi9*(t)*,* respectively (Figure 2d), which could be used to identify the accessions resistant to *P. oryzae* according to targeted genes. Among them, four isolates, TH20, JL19, LY23, and YJ9, showed different reaction patterns between IRBLta2-*Pi* and IRBLta2-Re, the same resistance gene *Pita-2*. And, two isolates, CC13 and SY1, showed different reaction patterns between IRBLta-K1 and IRBLta-CP1, which harbored the same resistance genes *Pita* (Table 3). Furthermore, two strong virulence isolates, JL27 and SY5, were selected because they were pathogenic to all MDVs harboring all resistance genes (Table 3). The existence of super-virulent isolates indicated a severe threat to rice production. Meanwhile, super-virulent isolates could be used to digest some broad-spectrum resistant accessions to fill special shortcomings in the region.

## 4. Discussion

Based on the reaction patterns of 25 DVs harboring 23 resistance genes, backcross parent LTH, and check variety Menggudao, we identified 206 isolates from samples collected in three regions in Jilin Province, northeast China (Figure 1 and Table 1). Based on the criteria of the U-i-k-z-ta pattern, we clarified 175 races among 206 isolates, whereas race U73-i5-k177-z11-ta330 included 5 isolates, which were predominant among the races. Furthermore, among MDVs subgroups, U73(100 reaction types, 48.5%) and U53(44, 21.4%); i7(63, 30.6%), i5(65, 31.6%), and i1(45, 21.8%); k177(85, 41.3%); and ta330 (87, 42.2%) were predominant (Table 2). Thirty isolates demonstrated no virulence to LTH, suggesting that there were some resistance genes existing in the LTH variety, and the same events occurred in Kenya, where four isolates showed no virulence [17].

Among 25 MDVs and LTH, those harboring resistance genes *Pish*, *Pit*, *Pia*, *Pii*, *Pik-s*, *Pik*, *Pita*(two lines), and *Pita-2*(two lines) had the highest virulence frequency of *P. oryzae* ranging from 80 to 100%, followed by intermediate, 40–80%, for *Pib*, *Pi5*(t), *Pik-m*, *Pi1*, *Pik-h*, *Pik-p*, *Pi7*(t), *Piz*, *Piz-5*, and *Piz-t*, and low, 0–40%, for *Pi3*, *Pi9*(t), *Pi12*(t), *Pi19*(t), and *Pi20*(t). Compared with the previous 44 isolates studied, the resistance frequencies of *Pi9*(t), *Pi19*(t), *Piz*, *Piz-5*, *Piz-t*, *Pi12*(t), *Pi5*(t), and *Pik-h* were all more than 60% [15]. But, besides the *Pi9*(t), *Pi12*(t), *Pi19*(t), and *Pi3*, the *Pi20*(t) was more than 60% in this study, and the pathogenic characteristics of the isolates changed. The complex situation of the blast population might be related to the rice cultivars or the number of blast samples collected in different regions [17].

We divided the identified isolates into two groups (I and II) based on their reaction pattern on 25 MDVs. Those predominant races had high frequencies of reaction types. The frequency variation of blast isolates to DVs is essential for race diversity. For example, in race Subgroup Ib, the U63, the major difference from IRBLb-B(*Pib*), occurred twice in total, and the frequency was 66.7%. This was the major reason attributable to the Ib group. Similarly, the presence and absence of U53, z07, and k177 were major differences between Group I and Group II. Altogether, the diversity of all the major reaction types corresponded to the distribution of isolates of the cluster. The diversity index of DVs Groups z and k, which was calculated by the Simpson method [32], was higher than that of the other three groups. The diversity of avirulence genes for the blast isolates was important to differentiate the cluster groups.

Even though the set of MDVs has been used for pathogenic study in China, we were the first to use the U-i-k-z-ta criterion for naming each race of MDVs and screening SDBIs of blast fungus. Based on the gene-for-gene theory, every resistance gene in the host crop corresponds to an avirulence gene in the pathogen [3]. Wang et al. [15] were the first to inoculate 44 blast fungal isolates originated from Jilin to 23 monogenic line varieties. Isolate frequencies to MDVs of *Pi9*(t), *Pi19*(t), *Piz*, *Piz-5*, *Piz-t*, *Pi12*(t), *Pi5*(t), and *Pik-h* were 94.2%, 84.1%, 81.8%, 81.8%, 79.5%, 72.7%, 68.2%, and 68.2%, respectively, which clarified that *avrPi9*(t), *avrPi19*(t), *avrPiz*, *avrPiz-5*, *avrPiz-t*, *avrPi12*(t), *avrPi5*(t), and *avrPik-h* were the predominant avirulence gene types. In Hunan Province, China, Xing et al. [34] inoculated the same rice monogenic lines using their local isolates and demonstrated that *avrPi9*(t), *avrPiz-5*, *avrPik-h*, and *avrPik-m* were predominant with a frequency of 91.6%, 91%, 87.9%, and 87.3%, respectively. In Chongqing, China, Li [35] reported that *avrPik-m* and *avrPik-h* were the predominant avirulence genes, both frequency with 82.18%, followed by *avrPi9*(t), *avrPik,* and *avrPi7*(t) with 76.24%, 73.27%, and 70.30%, respectively. In Heilongjiang Province, China, Zhang et al. [36] found that *avrPi12*(t), *avrPi9*(t), *avrPi19*(t), *avrPi20*(t), and *avrPiz-t* had high frequency with 74.69%, 72.28%, 68.67%, 68.07%, and 60.24%, respectively. Altogether, isolate frequency to the DVs contained *Pi9*(t) is low, but to the DVs contained *Pi12*(t) varies differently according to the areas, which suggests that *Pi9*(t) is the most stable resistance gene. The structure of a pathogen population is strongly influenced by the structure of its host population [37]. The complex situation of the blast population might be related to the rice cultivars that are cultivated in certain regions [17].

Also, this same set of MDVs have been used in many countries, including China [15], Western Africa [22], Cambodia [21], the United States [30], Japan [16], Bangladesh [19], Kenya [17], Vietnam [20], and Indonesia [18]. This set of MDVs in our study could be an effective tool to clarify the pathogenic characteristics of blast fungus, search for resistance germplasm, and develop cultivars resistant to blast worldwide. The virulences of isolates to some MDVs are similar across many countries, such as the MDVs harboring *Pia*, *Pii,* and *Pit* are all high, but *Pi9*(t), *Piz*, and *Pib* are low in almost all the countries. However, the virulences of the MDVs harboring other resistance genes are different. For example, *Pi12*(t), *Pi19*(t), and *Pi20*(t) isolates have low virulence in Jilin Province, northeast China, but *Pi12*(t) and *Pi20*(t) isolates have similar virulence in northeast China and Japan [16]. But, *Pi19*(t) and *Pi20*(t) isolates have higher virulence frequency in Bangladesh, Kenya, and Cambodia than in other countries. In Indonesia, *Pi12*(t) isolates have high virulence, but *Pi19*(t) and *Pi20*(t) isolates have moderate virulence. *Pita* (two lines) and *Pita-2* (two lines) isolates have higher virulence frequency and greater differentiation in Cambodia than in other Asian countries (Appendix A; Appendix A).

We selected 27 SDBIs based on the distinct non-pathogenic reaction patterns corresponding to 25 MDVs (Table 3). These SDBIs could be used as the basic tool to evaluate rice germplasm for resistance to blast. Assisted with those strong virulent isolates such as JL27 and SY5, we will mine novel resistance genes to blast. We identified some virulent isolates with high frequency, which may break the existing resistance and bring about an epidemic of blast in the *Japonica* rice-planting area. Those SDBIs could be used to develop a durable system against blast disease in the *Japonica* area in northeast China, and the isolates could be replaced according to the modification abilities or virulence stability. Furthermore, the designation system for 25 MDVs in our study could be used to compare blast races in different regions and countries so as to help understand the differentiation and diversity of blast races in the world [16,17,18,19,20,21,22,30,31].

## 5. Conclusions

Using the U-i-k-z-ta pattern criteria, we first named 175 race types among 206 virulent isolates of *P. oryzae* in China. A total of 27 SDBIs were selected, and the set of strains would help us to develop a durable system against rice blast disease in northeast China. This study described the pathogenic diversity of blast isolates in the *Japonica* rice area of northeast China and compared the pathogenic diversity of isolates with other countries, which could help us to develop cultivars resistant to blast around the world.

## Figures and Tables

**Figure 1 pathogens-13-00211-f001:**
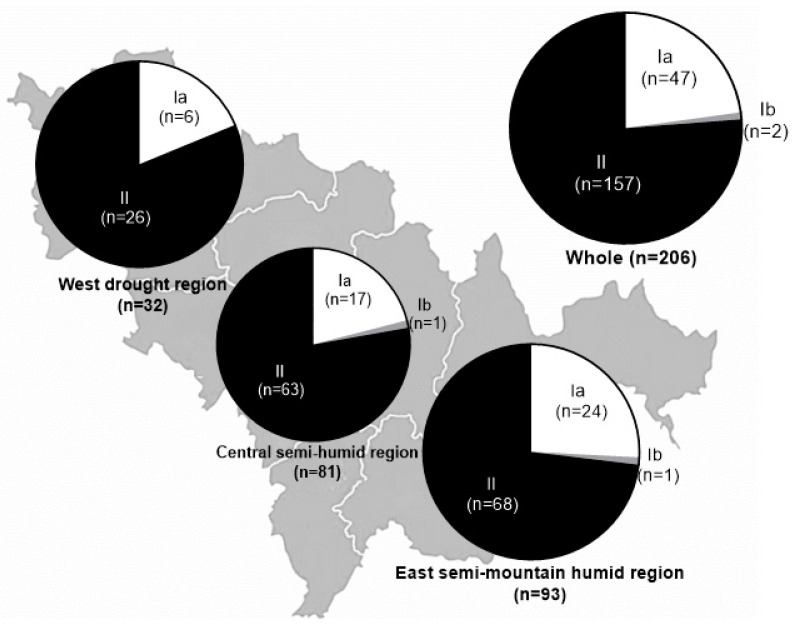
Distribution of blast isolates identified from three regions in Jilin Province, northeast China.

**Figure 2 pathogens-13-00211-f002:**
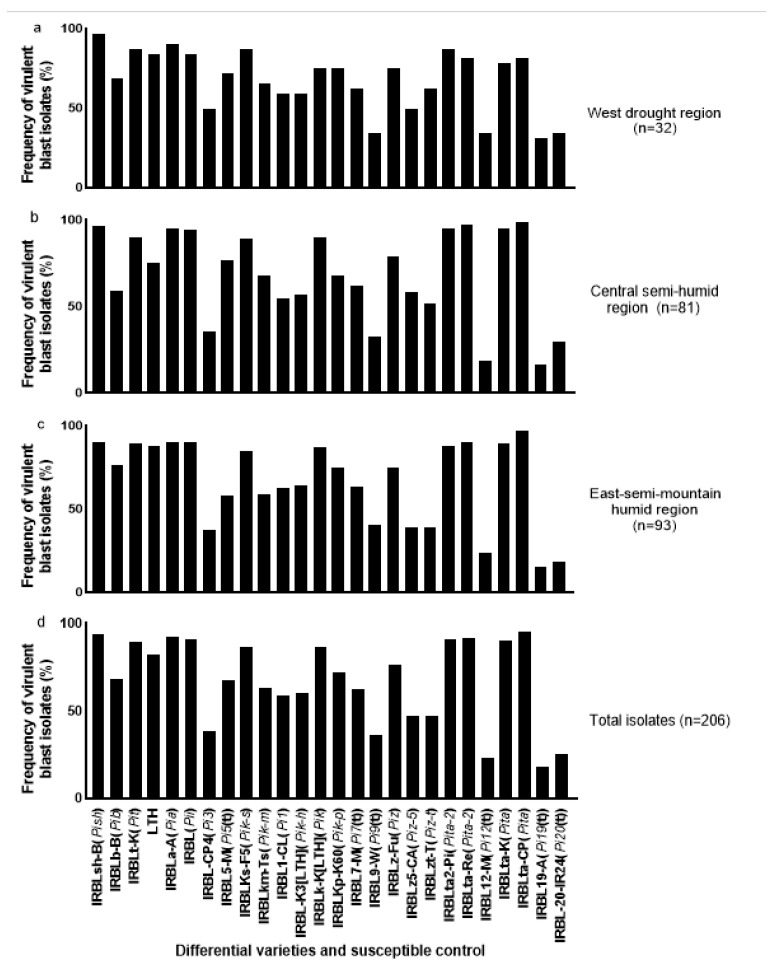
Pathogenic frequencies of blast isolates to differential varieties (DVs) in *Japonica* rice samples collected from three regions in Jilin Province, northeast China. (**a**) frequency of virulent blast isolates in west drought region; (**b**) frequency of virulent blast isolates in central semi-humid region; (**c**) frequency of virulent blast isolates in east-semi-mountain humid region; (**d**) frequency of virulent blast isolates of total isolates.

**Table 2 pathogens-13-00211-t002:** Reaction types of differential varieties (DVs) in each subgroup.

Reaction to Resistance Gene in Differential Variety										Cluster Group				
Reaction type	*Pish*	*Pib*	*Pit*	LTH	*Pia*	-	-	-	-	Ia (n = 47)	Ib (n = 2)	II (n = 157)	Total (n = 206)	
U73	v	v	v	v	v	-	-	-	-	39	0	61	100	48.5
U63	a	v	v	v	v	-	-	-	-	0	2	1	3	1.5
U53	v	a	v	v	v	-	-	-	-	0	0	44	44	21.4
Others	U72, U71, U70, U62, U61, U52, U43, U40, U33, U32, U31, U23, U22, U21, U13, U11, U03, U01									8	0	51	59	28.6
Diversity index										0.31	0.00	0.76	0.71	
Reaction type	*Pii*	*Pi3*	*Pi5*(t)	-	-	-	-	-	-					
i7	v	v	v	-	-	-	-	-	-	2	0	61	63	30.6
i5	v	a	v	-	-	-	-	-	-	15	0	50	65	31.6
i1	v	a	a	-	-	-	-	-	-	22	2	21	45	21.8
Others	i6, i4, i3, i2, i0									8	0	25	33	16
Diversity index										0.68	0.00	0.73	0.76	
Reaction type	*Pik-s*	*-*	*-*	*Pik-m*	*Pi1*	*Pik-h*	*Pik*	*Pik-p*	*Pi7*(t)					
k177	v	-	-	v	v	v	v	v	v	0	0	85	85	41.3
k101	v	-	-	a	a	a	v	a	a	15	0	6	21	10.2
Others	k176, k175, k174, k173, k171, k167, k165, k163, k157, k156, k154, k147, k145, k143, k137, k131, k127, k123, k121, k117, k114, k113, k112, k111, k107, k106, k103, k102, k100, k077, k067, k057, k055, k047, k032, k020, k017, k012, k010, k003, k002, k001, k000									32	2	66	100	48.5
Diversity index										0.87	1.00	0.70	0.81	
Reaction type	*Pi9*(t)	*-*	*-*	*Piz*	*Piz-5*	*Piz-t*	-	-	-					
z17	v	-	-	v	v	v	-	-	-	3	0	37	40	19.4
z07	a	-	-	v	v	v	-	-	-	0	0	30	30	14.6
z01	a	-	-	v	a	a	-	-	-	15	1	28	44	21.4
z00	a	-	-	a	a	a	-	-	-	21	0	6	27	13.11
Others	z16, z15, z14, z13, z12, z11, z10, z06, z05, z04, z03, z02									8	1	56	65	31.55
Diversity index										0.70	1.00	0.86	0.87	
Reaction type	*Pita-2* (Pi)	*Pita-2* (Re)	*Pi12*(t)	*Pita* (K1)	*Pita* (CP1)	-	*Pi19*(t)	*Pi20*(t)	-					
ta332	v	v	a	v	v	-	a	v	-	2	0	21	23	11.2
ta330	v	v	a	v	v	-	a	a	-	25	2	60	87	42.2
Others	ta733, ta732, ta731, ta730, ta713, ta620, ta431, ta333, ta331, ta322, ta320, ta312, ta300, ta233, ta232, ta230, ta221, ta220, ta132, ta130, ta120, ta031, ta030, ta021, ta010, ta000									20	0	76	96	0.47
Diversity index										0.71	0.00	0.82	0.79	

**Note:** a, avirulent; v, virulent.

**Table 3 pathogens-13-00211-t003:** Standard differential blast isolates (SDBIs) collected in three regions of Jilin Province.

Monogenic Lines Varieties Harboring Resistance Genes																										
Isolate Code	IRBLsh-B(*Pish*)	IRBLb-B(*Pib*)	IRBLt-K59(*Pit*)	IRBLa-A(*Pia*)	IRBL-F5(*Pii*)	IRBL3-CP4(*Pi3*)	IRBL5-M[*Pi5*(t)]	IRBLks-F5(*Pik-s*)	IRBLkm-Ts(*Pik-m*)	IRBL1-CL(*Pi1*)	IRBLkh-K3[LT](*Pik-h*)	IRBLk-k[LT](*Pik*)	IRBLkp-K60(*Pik-p*)	IRBL7-M[*Pi7*(t)	IRBL9-W[*Pi9*]	IRBLz-Fu(*Piz*)	IRBLz5—CA(*Piz-5*)	IRBLzt-T(*Piz*-t)	IRBLta2-Pi(*Pita-2*)	IRBLta2-Re(*Pita-2*)	IRBL12-M[*Pi12*(t)]	IRBLta-K1(*Pita*)	IRBLta-CP1(*Pita*)	IRBL19-A[*Pi19*]	IRBL20-IR24[*Pi20*(t)]	LTH
CC24	a	v	v	a	v	a	v	v	v	v	a	v	v	v	v	v	v	v	v	v	v	v	v	a	v	v
SY4	v	a	v	v	v	v	v	v	v	v	v	v	v	v	v	v	v	v	v	v	v	v	v	v	v	v
YJ33	v	v	a	v	v	v	a	v	v	a	v	v	v	v	v	v	v	v	v	v	v	v	v	a	a	v
CC31	a	v	v	a	v	v	v	v	v	v	v	v	v	v	v	v	v	v	v	v	a	v	v	a	a	v
LY7	v	a	v	v	a	v	v	a	v	v	v	v	v	v	v	v	v	v	a	v	a	v	v	a	a	v
SY7	v	a	v	v	v	a	v	v	v	v	v	v	v	v	v	v	v	v	v	v	v	v	v	v	v	v
SY12	v	v	v	v	v	v	a	v	v	v	v	v	v	v	a	v	v	v	v	v	a	v	v	a	a	a
YJ23	v	a	v	v	v	v	v	a	v	v	v	v	v	v	v	v	v	v	v	v	v	v	v	v	a	v
TH20	v	v	v	v	v	a	a	v	a	v	v	v	v	v	a	v	v	v	a	v	a	v	v	a	a	a
JL19	v	a	v	v	v	a	v	v	v	a	v	v	v	v	a	v	v	v	v	v	a	v	v	a	a	v
CC12	v	a	v	v	a	v	v	v	v	v	a	v	v	v	a	a	v	v	v	v	a	v	v	a	a	v
YJ15	v	a	v	v	v	a	v	v	v	v	v	a	a	v	a	v	a	a	v	v	a	v	v	a	a	v
YJ39	v	v	v	v	v	v	v	v	a	v	v	v	a	v	a	v	a	a	v	v	v	v	v	v	a	v
TH19	v	v	v	v	v	v	v	v	v	v	v	v	v	a	a	v	v	v	v	v	a	v	v	a	v	v
SY15	v	v	v	v	v	v	v	v	v	v	v	v	v	v	a	v	v	v	v	v	v	v	v	v	v	v
YJ31	v	v	a	v	v	v	a	v	v	v	v	v	v	v	v	a	a	v	v	v	v	v	v	a	a	v
YJ14	v	v	v	v	v	v	v	v	v	v	v	v	v	v	v	v	a	v	v	v	a	v	v	v	a	v
TH10	v	v	a	v	v	a	v	v	v	v	v	v	v	v	v	v	v	a	v	v	v	v	v	a	a	v
LY23	v	v	v	v	v	v	v	v	v	v	v	v	v	v	v	v	v	v	a	a	v	v	v	v	a	v
YJ9	v	v	v	v	v	v	v	v	v	v	v	v	v	v	v	v	v	a	v	a	a	a	v	a	a	v
SP11	v	v	v	v	v	v	v	v	v	v	v	v	v	v	v	v	v	v	v	v	a	v	v	v	v	v
CC13	v	a	v	v	v	a	v	v	v	v	a	v	a	a	a	v	v	v	a	v	v	a	v	a	a	v
SY1	v	v	v	v	v	a	v	a	v	v	a	a	v	a	a	a	v	a	v	v	v	v	a	v	v	v
LY1	v	v	v	v	v	v	v	v	v	v	v	v	v	v	v	v	v	v	v	v	v	v	v	a	v	v
TH11	v	v	v	v	v	v	v	v	v	v	v	v	v	v	v	v	v	v	v	v	v	v	v	v	a	v
JL27	v	v	v	v	v	v	v	v	v	v	v	v	v	v	v	v	v	v	v	v	v	v	v	v	v	v
SY5	v	v	v	v	v	v	v	v	v	v	v	v	v	v	v	v	v	v	v	v	v	v	v	v	v	v

**Note:** CC is the abbreviation of Chang-Chun; SY, Song-Yuan; TH, Tong-Hua; JL, Ji-Lin; SP, Si-Ping; LY, Liao-Yuan; YJ, Yan-Ji; v, virulent; a, avirulent.

## Data Availability

All the data generated during the current study are included in the manuscript.

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
