# Peer review of "Pathogenicity Analyses of Rice Blast Fungus (Pyricularia oryzae) from Japonica Rice Area of Northeast China"

_pathogens, 2024, doi:10.3390/pathogens13030211_

Round 1

Reviewer 1 Report

Comments and Suggestions for Authors

Wang et al. presented important works regarding rice blast fungus and pathogenicity patterns in northeast China. The experimental design requires labor intensive plant pathology related works, and the authors did a good job. Still, I would like to make some points clear.

1. I can't find the supplementary page and no access to the dendrogram and other information. This should be fixed.

2. Regarding dendrogram, the authors should provide in the materials and methods section how they did.

3. The authors should provide more background knowledge regarding the system they are studying. For example, what is the U-i-k-z-ta pattern you applied here?

4. The authors used 1-5 scale (later on being transferred to the binary system). Although they cited what the numbers mean, it would be better to provide the details about each score.

5. The authors should provide how they obtained the plant differential varieties. 

6. Line 241-242- IRBLta-K1 and IRBLta-CP1- same R gene, but different response- the authors need to write their speculation why it is happening. 

7. Figure 2 and Table 3 seem to have a lot of redundant information. Why is Table 3 so special as Figure 2 already contained lots of info in Table 3?

Comments on the Quality of English Language

Discussion part should be carefully checked. 

Author Response

Dear reviewer,

Thanks for your constructive suggestions to our Manuscript. We revised the MS step by step according to your suggestions. Please grant instruction If there were problems.

  1. I can't find the supplementary page and no access to the dendrogram and other information. This should be fixed.

Answer: We have added the supplementary including the dendrogram and other information.

  1. Dendrogram section how they did.

Answer: We have supplied the the materials and methods regarding dendrogram.

  1. The authors should provide more background knowledge regarding the system they are studying. For example, what is the U-i-k-z-ta pattern you applied here?

Answer: We have supplied some background knowledge in text.

Seeds of the monogenic lines were generously provided by Cailin Lei at the Institute of Crop Sciences, Chinese Academy of Agricultural Sciences (ICS-CAAS), the LTH is a Japonica-type local variety from Yunnan Province, China, that is broadly susceptible to rice blast isolates from the Jilin Province.

Five seeds of each MDVs and CK were planted in plastic tray (68 × 34 × 7 cm) containing local clay soil which had been treated with 0.1% Carbendazim fungicide for 3 d. The tray was kept at 20℃ for night and 28℃ for day for about four weeks until seedlings were at the 4th - 5th leaf stage.

About the U-i-i-z-ta pattern were noted according to reference article. If the reader was interested it, he could find the details of the method as we noted. If we added it in main body, it would occupy much space in main body. An they are listed as following.

These DVs and LTH were divided into five groups: U, i, k, z and ta. Group U includes LTH and four DVs(IRBLa-A for Pish, IRBLb-B for Pib, IRBLt-k59 for Pii, IRBL3-CP4 for Pi3, and IRBL5-M for Pi5(t)) with the Pii locus on chromosome 9; group k includes seven DVs (IRBLks-F5 for Pik-s, IRBLk-m, IRBL1-CL for Pi1, IRBLkh-k3[LT] for Pik-h, IRBLk-K[LT] for Pik, IRBLkp-K60 for Pik-p, and IRBL7-M for Pi7(t) with the Pik region on chromosome 11; group z includes four DVs(IRBL9-W for Pi9(t), IRBLz-Fu for Piz, IRBLz5-CA for Piz-5, and IRBLzt-T for Piz-t) with the Piz region on chromosome 6; and group ta includes seven DVs (IRBLta2-Pi for Pita-2, IRBLta-Re for Pita-2, IRBL12-M for Pi12(t), IRBLta-K1 for Pita, IRBLta-CP1 for Pita, IRBL19A for Pi19(t), and IRBL20-IR24 for Pi20(t)) with the Pita region on chromosome 12. The 21resistance genes in group I, k, z, and ta are multiple alleles or are located in the same chromosomal regions. The other 4 genes (Pia on chromosome 11, Pish and Pit on chromosome 1, and Pib on chromosome 2 are located independently in different chromosomal regions and are gathered with LTH in group U. The DVs in each of the five groups u, i, k, z and ta were divided into one to three subunits, with one to three DVs in each subunit. To each of the DVs within each subunit, the number 1, 2 and 4 were given if the respective DV showed a susceptible reaction to a blast isolate(i.e., the code 1 was assigned if the first DV in the subunit was susceptible, 2 was assigned if the second DV in the subunit was susceptible, and was assigned if the third DV in the submit was susceptible). P.oryzae races were designated by the combined sum of each subunit. Isolates classified this way were designated as reaction type with each DV group and as races based on the set of all five reactions type.

  1. The authors used 1-5 scale (later on being transferred to the binary system). Although they cited what the numbers mean, it would be better to provide the details about each score.

Answers: We have added the description of lesion types about each score.

     Blast reaction was classified on a scale of 0-5 with 0 means no evidence of infection; 1 means brown specks smaller than 0.5 mm, no sporulation; 2 means brown specks about 0.5-1.0 mm in diameter, no sporulation; 3 means roundish to elliptical lesion about 1-3 mm in diameter with a gray center surround by brown margins, lesions capable of sporulation; 4 means typical spindle-shaped lesions capable of sporulation, longer than 3 mm with necrotic gray centers and water-soaked or reddish brown margins, little or no coalescence of lesions; and 5 means the same as type 4 but with half of one or two leaves blades killed by the coalescence of lesions. as described by Hayashi and Fukuta [31] using Menggudao as susceptible check.

  1. The authors should provide how they obtained the plant differential varieties. 

Answers: We have offered the information about the original source in text.

Seeds of the monogenic lines and LTH were generously provided by Cailin Lei at the Institute of Crop Sciences, Chinese Academy of Agricultural Sciences (ICS-CAAS), LTH is a Japonica-type local variety from Yunnan Province, China, that is broadly susceptible to rice blast isolates from the Jilin Province, China.

  1. Line 241-242- IRBLta-K1 and IRBLta-CP1- same R gene, but different response- the authors need to write their speculation why it is happening. 

Answer: We have discussed the cause. Because the genetic background were differential, so even the same R gene, it presents different phenotype .

  1. Figure 2 and Table 3 seem to have a lot of redundant information. Why is Table 3 so special as Figure 2 already contained lots of info in Table 3?

Figure 2 focuses on Pathogenic frequencies of blast isolates to differential varieties (DVs) in Japonica rice samples collected from three regions in Jilin province, northeast China, and table 3 presented Standard differential blast isolates (SDBIs) collected in three regions of Jilin province. Even Figure 2 and Table 3, the presented information could make clear about the context.

Last but not least, thanks very much for your dedication to my manuscript.

Best regards,

All authors of this MS

Reviewer 2 Report

Comments and Suggestions for Authors

The manuscript entitled “Pathogenicity analyses of rice blast fungus (Pyricularia oryzae) from Japonica rice area of northeast China” is a very interesting study with with results of great relevance for rice cultivation. Some details are recommended for the authors:

1. The authors are recommended to standardize the description of the LTH variety throughout the entire manuscript and include the Menguguado variety in all sections where rice varieties are described, along with specifying its role.

2. The authors are advised to include the mentioned supplementary material, as it is not currently present in the file, and much of the information described in the manuscript is found there.

3. Similarly, the authors are encouraged to ensure consistency in describing the procedure for obtaining the 206 isolates across the Materials and Methods, Results, and Discussion sections.

4. The authors are also recommended to enhance the description of the methodology in Section 2.3. For instance, how many plants of each rice variety were used in total to evaluate each isolate? Under what physical and environmental conditions were the evaluations conducted? For how long? And so on.

5. Lastly, it is suggested that the authors provide a bit more detail on how the grouping of isolates into groups Ia, Ib, and II was carried out.

Author Response

Dear reviewer,

Thanks for your constructive suggestions to our Manuscript. We revised the MS step by step according to your suggestions. Please grant instruction If there were problems.

  1. The authors are recommended to standardize the description of the LTH variety throughout the entire manuscript and include the Menguguado variety in all sections where rice varieties are described, along with specifying its role.

Answer: We have offered the description of the LTH and Menggudao varieties.

The LTH is a Japonica-type local variety from Yunnan Province, China, that is broadly susceptible to rice blast isolates from the Jilin Province, which is the backcrossed parent of those monogenic lines DVs. The Menggudao is used as a super-susceptible variety as control or induce susceptible blast disease variety in disease nursery in China.

  1. The authors are advised to include the mentioned supplementary material, as it is not currently present in the file, and much of the information described in the manuscript is found there.

Answer: We have added the supplementary materials.

  1. Similarly, the authors are encouraged to ensure consistency in describing the procedure for obtaining the 206 isolates across the Materials and Methods, Results, and Discussion sections.

Answer: we have tried to offer some details as flowing. Such as disease scales of blast on rice leaves.

         Blast reaction was classified on a scale of 0-5 with 0=no evidence of infection; 1 = brown specks smaller than 0.5mm, no sporulation; 2 = brown specks about 0.5-1.0mm in diameter, no sporulation; 3 = roundish to elliptical lesion about 1-3 mm in diameter wit a gray center surround by brown margins, lesions capable of sporulation; 4 = typical spindle-shaped lesions capable of sporulation, longer than 3 mm with necrotic gray centers and water-soaked or reddish brown margins, little or no coalescence of lesions; and 5=same as type 4 but with half of one or two leaves blades killed by the coalescence of lesions.

     The LTH is a Japonica-type local variety from Yunnan Province, China, that is broadly susceptible to rice blast isolate. The Menggudao is used as a super-susceptible control variety or induce variety susceptible blast disease in disease nursery in China.

  1. The authors are also recommended to enhance the description of the methodology in Section 2.3. For instance, how many plants of each rice variety were used in total to evaluate each isolate? Under what physical and environmental conditions were the evaluations conducted? For how long? And so on.

Answer: We have added the details about inoculation condition.

The tray was kept at 20℃ for night and 28℃ for day for about four weeks until seedlings were at the 4th - 5th leaf stage. And after inoculation, the trays were kept in dark for 16 h before being transferred to greenhouse at 25-28℃ with 95% relative humidity.

Blast reaction was classified on a scale of 0-5 with 0 means no evidence of infection; 1 means brown specks smaller than 0.5 mm, no sporulation; 2 means brown specks about 0.5-1.0 mm in diameter, no sporulation; 3 means roundish to elliptical lesion about 1-3 mm in diameter with a gray center surround by brown margins, lesions capable of sporulation; 4 means typical spindle-shaped lesions capable of sporulation, longer than 3 mm with necrotic gray centers and water-soaked or reddish brown margins, little or no coalescence of lesions; and 5 means the same as type 4 but with half of one or two leaves blades killed by the coalescence of lesions.

  1. Lastly, it is suggested that the authors provide a bit more detail on how the grouping of isolates into groups Ia, Ib, and II was carried out.

Answer: We have supplied and addressed the information in line 138-140, line 260-270. They were based on the Supplementary Fig. S1.

     We divided the identified isolates into two groups (I and II) based on their reaction pattern on 25 MDVs. Those predominant races had high frequencies of reaction types. Frequency variation of blast isolates to DVs is essential for race diversity. For example, in race subgroup Ib, the U63, the major different from IRBLb-B(Pib) occurred twice in total and the frequency was 66.7%, this was the major reason attributable to Ib group. Similarly, the presence and absence of U53, z07 and k177 were a major difference between group I and group II. All together,the diversity of all the major reaction types was corresponding to the distribution of isolates of cluster. The diversity index of DVs group z and k which was calculated by Simpson method [32], which was higher than that of other 3 groups. The diversity of avirulent genes for the blast isolates was important to differentiate the cluster groups.

Last but not least, thanks very much for your dedication for my manuscript.

Best regards,

All authors of this MS

Reviewer 3 Report

Comments and Suggestions for Authors

Please check the attached file

Author Response

Dear reviewer,

Thanks very much for your pertinent suggestions.

This manuscript focuses on the pathogenicity of rice blast from Japonica area in northeast China. This is a key problem in rice breeding and production in world wide. Even using the modern molecular breeding technology, the design breeders firstly have to master the pathogenicity background of P. oryzae, then to hunt ideal donor which contained someone broad spectrum resistant gene(s) to polymerize into some one elite cultivar. At last, to distribute these cultivars according to the background both resistance from rice and pathogenicity from blast fungi.

This MS emphasized the blast fungi pathogenic in different regions or countries and compared their pathogenic characteristics, aim to exchange or introduce new target germplasms for directing rice breeding. We did not carry out the direction of hunting the avirulent genes in pathogen stains using molecular genetic methods such as by PCR, because they were less matched between avirulent gene marker(s) and non-pathogenicity of blast fungi out of lab, or the relation were more complex, a great deal of reports have confirmed this phenomenon. Our objective emphasized the non-pathogenicity information for offering rice blast resistance breeding and directing distribution of new cultivars. All above are the main reason we composed the MS.

According to the reviewer suggestions, we revised the MS and presented step by step as following details. There are also a number of significant comments on the work, which are given below. Please grant instruction If there were problems.

  1. Line 99: why and for what purpose, the authors used carbendazim to treat the soil if it has a systemic effect and can penetrate the seedling and to some extent inhibit the development of the disease. Please clarify in the text.

Answer: We ignored the common details and not do explain in text.

For avoiding others seeds borne disease pathogen to do harm to the seeds or seedling and affect the experiment, such as the fungi Fusarium Fujikuroi causes rice bakanae, we have to used carbendazim to treat the soil and seeds. We inoculated the seedling by spraying blast spores at rice 4-5 leaves in greenhouse, so there was less effect to disease development. On the contrary, if we did not, the results maybe be terrible.

  1. Line 108: a short explanation of the scale should be given.

Answer: We have added the scale information in text.

    Blast reaction was classified on a scale of 0-5 with 0 means no evidence of infection; 1 means brown specks smaller than 0.5 mm, no sporulation; 2 means brown specks about 0.5-1.0 mm in diameter, no sporulation; 3 means roundish to elliptical lesion about 1-3 mm in diameter with a gray center surround by brown margins, lesions capable of sporulation; 4 means typical spindle-shaped lesions capable of sporulation, longer than 3 mm with necrotic gray centers and water-soaked or reddish brown margins, little or no coalescence of lesions; and 5 means the same as type 4 but with half of one or two leaves blades killed by the coalescence of lesions. as described by Hayashi and Fukuta [31] using Menggudao as susceptible check.

  1. Section 2.5: it is unclear for what purpose the authors used “supervirulent” strains to screen varieties if they previously claimed that resistance was in a “gene-by-gene” system. In what proportions are“supervirulent”, low-virulent and medium-virulent strains common in real field conditions? Please add or discuss this point in the text.

Answers: a pertinent suggestion. We have added the explain and discuss this point in the text.

The rice cultivar lost its resistance to blast commonly because the pathogenic isolate occurred, and it overcame the R gene resistance, broken down the “gene for gene” system, it is the reason of the blast disease outbreak somewhere, usually. The proportions of super-virulent isolates were very low at initial stage but the new virulent strain would explore at the proper humidity and temperature condition on short time. So it is a prospective idea for screening resistant germplasms using super-virulent isolate(s). Furthermore, we selected some super-virulent isolates for screening super-resistant germplasm. 

  1. Section 2: it is unclear how the authors confirmed the relationship of the isolated isolates to the pathogen. As far as I can tell, this was done on the basis of symptoms, however, modern standards of phytopathology require the use of at least 2 methods, including one instrumental (for example, microscopy or PCR). It is also necessary to provide a detailed description of the strains, including morphological and genetic characteristics, which will certainly increase the value of the work. Also, according to the requirements accepted in phytopathology, it would be necessary to provide deposit numbers in Genbank or provide a link to articles where these strains were described earlier. Authors should clarify this point in the text.

Answer: Sorry, we omitted the description about morphological and genetic characteristics. We will do well according to the reviewer suggestion about pathogenic characteristic of some isolates in lasting research. Because the target of the MS focus on pathogenicity or virulence of isolates.

  1. Table 1: it is not entirely clear where we got 4 clusters from. Please clarify in the text.

Answer: We have supplied and addressed the information in line 138-140, line 260-270. They were based on the Supplementary Fig. S1.

     We divided the identified isolates into two groups (I and II) based on their reaction pattern on 25 MDVs. Those predominant races had high frequencies of reaction types. Frequency variation of blast isolates to DVs is essential for race diversity. For example, in race subgroup Ib, the U63, the major different from IRBLb-B(Pib) occurred twice in total and the frequency was 66.7%, this was the major reason attributable to Ib group. Similarly, the presence and absence of U53, z07 and k177 were a major difference between group I and group II. All together,the diversity of all the major reaction types was corresponding to the distribution of isolates of cluster. The diversity index of DVs group z and k which was calculated by Simpson method [32], which was higher than that of other 3 groups. The diversity of avirulent genes for the blast isolates was important to differentiate the cluster groups.

  1. Table 2: It is not entirely clear what 100% of the “frequency” value means. Authors should provide more precise information in the text.

Answer: the “100% of the “frequency” value” occurred in table 1. we have added this explain in table. The number total and frequency were matched.

  1. The Discussion section should be changed. This is due to the fact that for some reason the authors in this section provide data that is in the Results section, but this is not necessary.

Answer: Good idea. We have tried to revise.

  1. In addition, there is no critical analysis of the results obtained in comparison with work not in China, but in other countries. There is also a lack of information on follow-up plans for research prospects and the use of data obtained from the study.

Answer: A friendly suggestion. Because this is the first report using the ” U-i-k-z-ta” pattern criteria to analysis the pathogenicity characteristics in China, so the data only compared to countries.

We have supplied some plans for research prospects and the use of data obtained from the study: Assisted with those strong virulent isolates such as JL27 and SY5, we will mine novel resistant genes to blast. We identified some virulence isolates with high frequency, which may break the existing resistance and bring about an epidemic of blast in Japonica rice-planting area. Those SDBIs could be used to develop a durable system against blast disease in Japonica area in northeast China, and the isolates could be replaced according to the modification abilities or virulence stability.

Last but not least, thanks very much for your dedication for my manuscript.

Best regards,

All authors of this MS

Round 2

Reviewer 1 Report

Comments and Suggestions for Authors

I am pleased to inform you that the authors have taken into account all of my concerns from the previous version, and have made the necessary improvements. After reviewing the latest version, I can confidently say that it meets my expectations and requirements.

Author Response

Dear academic editor and reviewers,

      Thanks very much for your kindly suggestions and comments on the MS-(Pathogens-2816080) . We have revised the MS according to your comments point to point, as for the details, please see the attached files.

Best regards,

All the authors of MS-(Pathogens-2816080)

Reviewer 3 Report

Comments and Suggestions for Authors

All comments were taken into account by the authors in the text. I have no questions. I wish the authors further success

Author Response

Dear academic editor and reviewers,

      Thanks very much for your kindly suggestions and comments on the MS-(pathogens-2816080) . We have revised the MS according to your comments point to point, as for the details, please see the attached files.

Best regards,

All the authors of MS-(pathogens-2816080)